# Mechanism-Aware Prediction of Tissue-Specific Drug Activity via Multi-Modal Biological Graphs

**Sally Turutov** *turutovsally@campus.technion.ac.il*
*Technion - Israel Institute of Technology*

**Kira Radinsky** *kirar@cs.technion.ac.il*
*Technion - Israel Institute of Technology*

**Reviewed on OpenReview:** *https://openreview.net/forum?id=UDW8m9iQeC*

## Abstract

Predicting how small molecules behave across human tissues is essential for targeted therapy development. While some existing models incorporate tissue identity, they treat it as a label—ignoring the underlying biological mechanisms that differentiate tissues. We present Expresso, a multi-modal architecture that predicts tissue-contextual molecular activity as measured by the assay by modeling how compounds interact with transcriptomic and pathway-level tissue context. Expresso constructs heterogeneous graphs from GTEx data, linking samples, genes, and pathways to reflect expression profiles and curated biological relationships. These graphs are encoded using a hierarchical GNN and fused with frozen molecular embeddings to produce context-aware predictions. A multi-task pretraining strategy—spanning gene recovery, tissue classification, and pathway-level contrastive learning—guides the model to learn mechanistically grounded representations. On nine tissues, Expresso improves mean AUC by up to 27.9 points over molecule-only baselines. Our results demonstrate that incorporating biological structure—as defined by the assay—yields more accurate and interpretable models for tissue-specific drug behavior in human cell-based in vitro assay systems.

## 1 Introduction

While machine learning has advanced rapidly in molecular modeling, many state-of-the-art models still evaluate drug compounds in isolation—without considering the biological context in which those molecules operate. In reality, a compound's effect depends not just on its chemical structure but also on the specific tissue it enters, which genes are expressed there, and which biological pathways are active. These factors fundamentally shape how a drug behaves in the body. As a result, models that ignore biological context often struggle to generalize to new tissues or disease states, limiting their reliability in real-world applications.

Some recent approaches incorporate biological or cellular context alongside molecular information—for example, by embedding the identity of the cell line or by integrating transcriptomic profiles into the predictive model (Oskooei et al., 2018; Wu et al., 2025; Shahzad et al., 2024). While this provides useful context, these methods typically represent cellular state as a flat label or a static feature vector, without explicitly modeling the underlying regulatory programs, pathway structure, and gene–gene interactions that shape tissue-contextual responses measured by the assay. In the absence of such mechanistic structure, predictions may become brittle and miss critical determinants of tissue-specific drug behavior.

At the same time, large-scale transcriptomic datasets across human tissues have become increasingly accessible. These datasets don't merely label tissues—they characterize them, offering detailed, sample-level insight into gene activity and biological pathway dynamics. When properly integrated, this wealth of biological structure can serve as a powerful inductive bias for molecular prediction. Yet, most current architectures lack the capacity to exploit this information to generate biologically meaningful predictions.

We address this gap by introducing Expresso: *Expression and Pathway-driven Representation for Enhancing Sample-context and Specific Organ activity prediction.* Expresso is a biologically informed framework that integrates molecular embeddings with graph-based tissue representations built directly from gene expression and curated pathway networks. Instead of abstracting tissue context as a flat label, Expresso constructs a heterogeneous graph per tissue—where samples, genes, and pathways form a structured system. This graph encodes both expression magnitude and biological relationships, allowing the model to learn tissue-contextual activity as defined by the assay.

The Expresso architecture utilizes hierarchical tissue encoder that performs message passing over sample–gene–pathway graphs. This enables the model to learn rich tissue embeddings conditioned on both observed expression and known biological priors. During training, Expresso is optimized not only for molecular activity prediction but also for self-supervised tissue modeling—reconstructing masked gene expression, classifying tissue identity, and aligning pathway-level representations. This multi-objective setup strengthens generalization and interpretability of the model.

The contributions of this work are threefold: (1) We introduce Expresso, an architecture that integrates molecular structure with rich, expression-driven biological context by leveraging heterogeneous graphs composed of samples, genes, and pathways specific to each tissue—enabling accurate molecular activity prediction in complex biological environments. (2) We develop a novel graph-based tissue encoder, trained via a multi-objective learning framework, that effectively captures both the functional dynamics and structural organization of tissues by jointly modeling real gene expression data and curated pathway priors. (3) We provide new biological insights by identifying key pathways that mediate compound effects across diverse tissues, facilitating more interpretable and biologically grounded predictions.

By embedding molecular prediction in rich, tissue-specific biological context derived from human transcriptomic resources, Expresso moves toward more biologically faithful in-silico pipelines capable of modeling compound behavior with organ-level resolution as reflected in our in vitro assay panel. This enables more informed prioritization of candidate molecules based on their predicted tissue-specific activity in these assay-defined tissue contexts, potentially improving the efficiency and focus of subsequent experimental validation. We make our code and data openly available to the community for further study of the problem: `https://github.com/SallyTurutov/EXPRESSO`.

## 2 Related Work

Recent advances in molecular representation learning have produced powerful models for chemical property prediction. Approaches such as Mol-BERT (Li & Jiang, 2021), MolFormer (Wu et al., 2023), MMELON (Suryanarayanan et al., 2024), ChemBERTa (Chithrananda et al., 2020), and GROVER (Rong et al., 2020) learn rich, structure-aware representations from SMILES or molecular graphs, enabling strong performance on tasks such as solubility, toxicity, and general physicochemical property prediction. KA-GNNs (Bresson et al., 2025) further expand this capacity using heterogeneous graph structures with kernel-based attention to capture higher-order chemical dependencies. While effective for chemical-centric tasks, these methods operate on molecules in isolation and do not incorporate biological context, limiting them.

Several approaches attempt to include biological information alongside chemical representations. CLAMP (Seidl et al., 2023) aligns compounds with assay descriptions using contrastive learning, but the biological signal is limited to unstructured text. Domain-specific language models such as BioBERT (Lee et al., 2020) capture biomedical semantics for NLP tasks but are not designed to reason over molecular structures or tissue-specific gene expression. Pathformer (Liu et al., 2024) models gene–pathway and pathway–pathway relationships to improve interpretability in gene-level prediction, yet it does not support molecular input or tissue-specific expression data. These methods highlight the benefits of integrating biological structure, but they are not directly applicable to tissue-resolved molecular activity prediction.

Multimodal frameworks such as DeepCDR (Wu et al., 2022) and MultiDCP (Liu et al., 2020) combine molecular features with cell line-specific omics profiles to predict transcriptional perturbations or drug response. These approaches demonstrate that integrating chemical and cellular information can improve predictive performance.

Expresso unifies molecular encoders with structured biological knowledge. For each human tissue, Expresso leverages human cell-based assay data (predominantly established cell lines, with some primary cell systems where available), gene expression profiles, and pathway-level organization to create biologically informed embeddings that capture tissue-specific context as defined by these assays. This enables predictions of molecular activity that are sensitive to assay-specific tissue context, bridging part of the gap between purely chemical models and biologically informed in vitro tissue response.

## 3 Problem Definition

We address the problem of *tissue-specific molecular activity prediction*, where the goal is to determine whether a given molecule is active in a particular human tissue. This is framed as a binary classification task, where each input consists of a pair: a *molecule* and a *target tissue*. The model must whether the molecule exhibits tissue-contextual activity as measured in the assay.

Formally, each prediction instance is defined by: (1) A *molecule $m \in \mathcal{M}$*, represented by its SMILES string. (2) A *target tissue $t \in \mathcal{T}$*, from a set of human tissues. The model $f : \mathcal{M} \times \mathcal{T} \to \{0, 1\}$ is trained to predict whether $m$ is active in tissue $t$. Here, "activity" indicates whether a compound significantly affected a tissue-specific, biologically defined endpoint in a human cell-based assay—either primary cells or established human cell lines (e.g., cytokine release in lung cells or transcription factor activation in vascular cells); see Appendix A.2 for details.

To accurately model tissue-specific activity, the representation of each tissue must capture its full biological context. Rather than treating tissue as a categorical variable, we aim to build a learned embedding for each tissue that integrates multiple biological components. In particular, we leverage: (1) Samples: Each tissue is represented by multiple samples, where each sample corresponds to a gene expression profile derived from a different human donor. Each sample provides a vector of gene expression values (e.g., TPMs), giving insight into the transcriptional landscape of the tissue. (2) Genes: These are the measured units in each sample. Each gene has a quantitative expression level. The expression patterns across samples form the basis of the tissue's molecular signature. (3) Pathways: Biological pathways define sets of genes that participate in shared cellular processes or functions. By incorporating pathway membership information (e.g., a binary gene–pathway matrix), we can group genes into functionally meaningful modules that reflect regulatory and signaling mechanisms.

## 4 Model Architecture

We present Expresso, a biologically informed model that predicts molecular activity across human tissues by integrating chemical and transcriptomic data. It combines a molecular encoder with a hierarchical graph-based tissue encoder. Below, we detail the molecular encoder, graph construction and message passing, training losses, and the overall model architecture.

### 4.1 Molecular Encoder

To capture the structural properties of molecules, our model uses a pretrained molecular encoder that maps chemical inputs into continuous vector representations. Specifically, we adopt the compound encoder from CLAMP (Seidl et al., 2023), which has been trained on large-scale chemical–assay data using contrastive learning. This encoder processes molecular SMILES strings and generates fixed-dimensional embeddings that capture chemical features, which can be predictive of bioactivity when paired with biological context.

Formally, the encoder defines a mapping $f : \mathcal{M} \to \mathbb{R}^d$, where $\mathcal{M}$ denotes the space of molecules and $d$ is the embedding dimension. Given a molecule $m \in \mathcal{M}$, the output embedding is $\mathbf{m} = f(m)$. These embeddings serve as the molecular input to our downstream fusion and prediction modules.

Although we use CLAMP in our implementation, the architecture is modular and compatible with alternative molecular encoders. This design choice allows flexibility in adapting to different data settings or incorporating domain-specific encoders in future applications.

## 4.2 Tissue Graph Construction

To represent the complex biological landscape underlying tissue-specific molecular activity, we construct a heterogeneous graph that captures interactions across multiple biological levels: samples, genes, and pathways. This graph serves as the foundational structure for downstream tissue representation learning. Each tissue is modeled as an independent graph that integrates biological prior knowledge with tissue-specific measurements derived from gene expression data.

### 4.2.1 Node Types

The heterogeneous graph includes three distinct node types: (1) *Sample nodes* ($\mathcal{V}_s$), corresponding to individual biological samples. These nodes represent observed instances of gene expression within a tissue and preserve biological variability across samples. (2) *Gene nodes* ($\mathcal{V}_g$), corresponding to protein-coding genes measured in the transcriptomic profile. Each gene node encodes the role of a specific gene in mediating molecular function. (3) *Pathway nodes* ($\mathcal{V}_p$), representing curated biological pathways, which serve as biologically curated modules that organize genes into coherent functional units, enabling structured reasoning about cellular processes.

Let $N$, $G$, and $P$ denote the number of samples, genes, and pathways respectively in a given tissue graph. The full node set is thus $\mathcal{V} = \mathcal{V}_s \cup \mathcal{V}_g \cup \mathcal{V}_p$.

### 4.2.2 Edge Types

Edges in the graph encode functional and structural relationships across the biological hierarchy: (1) *Sample–Gene edges* ($\mathcal{E}_{s \leftrightarrow g}$), an undirected bipartite edge is formed between sample node $v_s \in \mathcal{V}_s$ and gene node $v_g \in \mathcal{V}_g$ if the gene $g$ has nonzero expression in that sample $s$. This relation captures the biological observation that samples express specific genes. The corresponding binary edge index is encoded as a sparse adjacency matrix $A_{sg} \in \{0,1\}^{N \times G}$, derived directly from the gene expression tensor $X \in \mathbb{R}^{N \times G}$ by thresholding nonzero entries. (2) *Gene–Pathway edges* ($\mathcal{E}_{g \leftrightarrow p}$) exist between gene nodes $v_g \in \mathcal{V}_g$ and pathway nodes $v_p \in \mathcal{V}_p$, defined according to a binary gene–pathway membership matrix $M \in \{0,1\}^{G \times P}$, where $M_{gp} = 1$ indicates that gene $g$ is known to participate in biological pathway $p$. This relation encodes structured biological prior knowledge from curated databases, capturing modular and interpretable groupings of genes that function together. These edges define a bipartite graph that links individual molecular measurements to higher-level biological functions. (3) *Pathway–Pathway edges* ($\mathcal{E}_{p \leftrightarrow p}$) model functional dependencies between biological processes by connecting pathway nodes $v_{p_1}, v_{p_2} \in \mathcal{V}_p$ via a pathway–pathway adjacency matrix $A_{pp} \in \{0,1\}^{P \times P}$, where $(A_{pp})_{p_1 p_2} = 1$ reflects known crosstalk or regulatory interactions between pathways $p_1$ and $p_2$. These edges capture relationships such as shared downstream targets, feedback loops, or co-activation, enabling information exchange between functionally related modules and supporting more holistic representations of tissue-specific context.

### 4.2.3 Node Features

Each node type is initialized with a biologically meaningful feature vector: (1) *Sample features* are computed from the raw gene expression profile. We first apply a simple transformation and normalization to obtain $\widetilde{X} \in \mathbb{R}^{N \times G}$. Then, each gene expression value is projected to a higher-dimensional space using a feed-forward network, producing sample-specific gene embeddings. These are aggregated across genes to produce a sample feature vector $h_s \in \mathbb{R}^d$ per node. (2) *Gene features* are initialized using the mean expression of each gene across samples. This results in a $G$-dimensional vector, further transformed using the same feed-forward encoder as above, yielding $h_g \in \mathbb{R}^d$. (3) *Pathway features* are derived from structural properties of the pathway–pathway network. We compute graph-theoretic statistics such as node degree and clustering coefficient, along with a spectral embedding of the adjacency matrix $A_{pp}$ (via Laplacian eigenmaps (Belkin & Niyogi, 2003)). These features are concatenated and standardized, producing a vector for each pathway, which is then projected into the GNN's hidden dimension: $h_p \in \mathbb{R}^d$.

#### 4.2.4 Biological Motivation

This graph formulation is biologically grounded and designed to capture the multi-scale organization of cellular function. By modeling *samples* explicitly as nodes, the graph preserves inter-sample variation, allowing the tissue representation to reflect within-tissue heterogeneity in gene expression. The inclusion of *gene* and *pathway* nodes enables mechanistic information flow through curated biological structures, supporting both interpretability and generalization. Furthermore, pathway–pathway edges encode regulatory coordination between biological processes, allowing the model to reason at a systems level. Altogether, the constructed heterogeneous graph $\mathcal{G} = (\mathcal{V}, \mathcal{E})$ provides a biologically meaningful representation of tissue context, which is then processed by a multi-layer heterogeneous GNN to produce compact, informative tissue embeddings for molecular activity prediction.

### 4.3 Graph Message Passing

To transform the constructed heterogeneous graph $\mathcal{G} = (\mathcal{V}, \mathcal{E})$ into a compact, informative tissue embedding, we employ a hierarchical graph neural network (GNN) architecture that iteratively refines node representations through multi-relational message passing.

#### 4.3.1 Initial Node Embeddings

For each node type, initial feature vectors are defined as follows: (1) *Sample nodes* $v_s \in \mathcal{V}_s$ are assigned features derived by projecting normalized gene expression profiles $X \in \mathbb{R}^{N \times G}$ through a feed-forward encoder, aggregating per-gene embeddings to form a sample representation $h_s^{(0)} \in \mathbb{R}^d$. (2) *Gene nodes* $v_g \in \mathcal{V}_g$ are initialized using the mean expression across samples, transformed via the same encoder, yielding $h_g^{(0)} \in \mathbb{R}^d$. (3) Pathway nodes $v_p \in \mathcal{V}_p$ encode structural graph properties of the pathway–pathway network using concatenated graph-theoretic features and spectral embeddings, linearly projected to obtain $h_p^{(0)} \in \mathbb{R}^d$.

#### 4.3.2 Layer-wise Message Passing

At GNN layer $\ell$, each node $v$ updates its embedding by aggregating information from neighbors of different types connected via specific edge relations $r \in \mathcal{R}$. Formally, for node $v$:

$$h_v^{(l+1)} = \text{Norm}\left( h_v^{(l)} + \sum_{r \in \mathcal{R}} \sum_{u \in \mathcal{N}_r(v)} \frac{1}{|\mathcal{N}_r(v)|} \cdot \text{Conv}_r^{(l)}\left(h_u^{(l)}\right) \right)$$

Here, $\text{Conv}_r^{(l)}$ denotes a relation-specific graph convolution layer (e.g., SAGEConv (Hamilton et al., 2017) or GATv2Conv (Brody et al., 2022)) tailored to edge type $r$, $\mathcal{N}_r(v)$ the set of neighbors of $v$ under relation $r$, and Norm a layer normalization operation. Residual connections ensure stability and mitigate over-smoothing across layers.

This multi-relation propagation mechanism integrates signals bidirectionally between samples and genes, between genes and pathways, and among pathways, thereby capturing local molecular activity, functional grouping, and higher-order regulatory crosstalk.

#### 4.3.3 Tissue-level Embedding Aggregation

After $L$ such layers, sample node embeddings $\{h_s^{(L)}\}$ represent context-aware molecular profiles within the tissue. To produce a fixed-length tissue embedding $\mathbf{z} \in \mathbb{R}^d$, these sample embeddings are pooled via a multi-query attention mechanism:

$$\mathbf{z} = \text{MultiQueryPooling}\left(\{h_s^{(L)}\}_{s \in \mathcal{V}_s}\right)$$

This pooling adaptively weights samples' contributions through learned attention queries, allowing the model to capture both dominant and subtle expression patterns relevant for tissue identity and function. Concretely,

we parameterize $Q$ learnable queries, each of which applies standard multi-head attention (Vaswani et al., 2017) over the set of sample embeddings. The resulting $Q$ context vectors are averaged to obtain $\mathbf{z}$.

Overall, this hierarchical GNN encoding framework enables biologically informed integration of heterogeneous molecular and functional information at multiple scales. By explicitly modeling samples, genes, and pathways with distinct features and relations, and leveraging multi-relational convolutions plus attentive pooling, the model learns rich, interpretable tissue representations that reflect both intra-tissue heterogeneity and systemic biological organization.

### 4.4   Multi-Task Loss Functions

To ensure that the learned tissue embeddings capture meaningful biological signals and generalize well for downstream molecular activity prediction, we train the model using a combination of complementary loss functions. These losses jointly encourage accurate reconstruction of molecular measurements, tissue classification, and robust representation of pathway-level functional modules.

#### 4.4.1   Gene Expression Reconstruction Loss

The heterogeneous graph explicitly models gene expression through sample–gene edges, enabling self-supervised learning by masking a fraction of gene expression values and tasking the model with their reconstruction. Let $X \in \mathbb{R}^{N \times G}$ denote the normalized gene expression matrix, and $\mathcal{M} \subseteq \{1, \ldots, N\} \times \{1, \ldots, G\}$ denote the set of sample–gene index pairs whose expression values are masked during training, randomly sampled across all samples and genes according to a fixed masking ratio. The model predicts masked expression values $\hat{X}_{i,j}$ via the inner product of learned sample and gene embeddings:

$$\hat{X}_{i,j} = h_{s_i}^{\top} h_{g_j}, \quad (i,j) \in \mathcal{M}$$

The reconstruction loss is defined as mean squared error over masked entries:

$$\mathcal{L}_{\text{gene}} = \frac{1}{|\mathcal{M}|} \sum_{(i,j) \in \mathcal{M}} \left( X_{i,j} - \hat{X}_{i,j} \right)^2$$

This loss encourages the model to learn embeddings that preserve detailed gene expression patterns, facilitating recovery of missing molecular signals.

#### 4.4.2   Tissue Classification Loss

To encourage tissue-specific features in the embedding space, we supervise the model to classify tissue identity from the aggregated tissue embedding $\mathbf{z}$. Given $C$ tissue classes and a ground-truth label $y \in \{1, \ldots, C\}$, a linear classifier $f_{\text{cls}}$ predicts logits $\ell \in \mathbb{R}^C$:

$$\ell = f_{\text{cls}}(\mathbf{z})$$

The tissue classification loss uses cross-entropy:

$$\mathcal{L}_{\text{cls}} = -\log \frac{\exp(\ell_y)}{\sum_{c=1}^{C} \exp(\ell_c)}$$

This supervision guides the tissue embedding to capture discriminative features relevant to tissue identity.

#### 4.4.3   Pathway Contrastive Loss

To promote pathway embeddings that reflect known functional relationships and are robust to noise, we apply a contrastive loss on two stochastic embeddings of each pathway node obtained via independent passes

through the GNN. For each pathway node $p$, let $\mathbf{z}_p^{(1)}, \mathbf{z}_p^{(2)} \in \mathbb{R}^d$ be its embeddings from these independent stochastic passes. Positive pairs $(\mathbf{z}_p^{(1)}, \mathbf{z}_p^{(2)})$ are contrasted against negative pairs with other pathway embeddings within the batch.

Using cosine similarity $\mathrm{sim}(\cdot, \cdot)$, the loss for pathway $p$ is:

$$\ell_p = -\log \frac{\exp\left(\mathrm{sim}(\mathbf{z}_p^{(1)}, \mathbf{z}_p^{(2)})/\tau\right)}{\sum_{q=1}^{P} \exp\left(\mathrm{sim}(\mathbf{z}_p^{(1)}, \mathbf{z}_q^{(2)})/\tau\right)}$$

where $\tau > 0$ is a temperature hyperparameter. The total pathway contrastive loss is averaged over all pathways:

$$\mathcal{L}_{\mathrm{pathway}} = \frac{1}{P} \sum_{p=1}^{P} \ell_p$$

This design leverages inherent stochasticity of the model to generate informative positive pairs, enabling robust and functionally meaningful pathway embeddings without requiring explicit graph augmentations.

### 4.4.4 Activity Prediction Loss

The primary supervised objective is molecular activity prediction. Given a molecular representation and a tissue embedding, the model predicts a scalar activity score. Let $\mathbf{m}_i$ denote the molecular embedding obtained from the molecular encoder, and $\mathbf{t}_i$ the tissue embedding produced by the tissue encoder for the $i$-th input. These are linearly projected and concatenated to form a joint representation:

$$\mathbf{z}_i = [\mathbf{W}_m \mathbf{m}_i \,\|\, \mathbf{W}_t \mathbf{t}_i],$$

where $\mathbf{W}_m$ and $\mathbf{W}_t$ are learned projection matrices, and $[\cdot \,\|\, \cdot]$ denotes vector concatenation. A classifier then maps $\mathbf{z}_i$ to a logit $\hat{y}_i \in \mathbb{R}$, representing the predicted activity score.

Given the ground-truth binary label $y_i \in \{0, 1\}$, we minimize a class-weighted binary cross-entropy loss:

$$\mathcal{L}_{\mathrm{activity}} = -\frac{1}{B} \sum_{i=1}^{B} \left[ \alpha \, y_i \log \sigma(\hat{y}_i) + (1 - y_i) \log(1 - \sigma(\hat{y}_i)) \right],$$

where $B$ is the batch size, $\sigma(\cdot)$ is the sigmoid function, and $\alpha = \frac{N_{\mathrm{neg}}}{N_{\mathrm{pos}} + \epsilon}$ is the positive-class weight used to mitigate class imbalance. This formulation balances contributions from active and inactive compounds, improving learning stability across tissues with skewed activity distributions. Overall, this loss directly supervises the model to distinguish active from inactive molecular responses in a tissue-specific context.

### 4.4.5 Overall Objective

The full training objective is a weighted sum of these losses:

$$\mathcal{L} = \mathcal{L}_{\mathrm{activity}} + \lambda_{\mathrm{gene}} \mathcal{L}_{\mathrm{gene}} + \lambda_{\mathrm{cls}} \mathcal{L}_{\mathrm{cls}} + \lambda_{\mathrm{pathway}} \mathcal{L}_{\mathrm{pathway}},$$

where each $\lambda$ is a tunable hyperparameter controlling the influence of the corresponding auxiliary task. This multi-task formulation enables the model to leverage both labeled activity data and structured biological information, encouraging tissue embeddings to reflect gene-level variation, tissue identity, and pathway-level organization—all while remaining optimized for the end predictive task.

### 4.5 Model Overview and Training

The full model consists of two main components: (1) a molecular encoder, and (2) a tissue encoder that constructs graph-based tissue embeddings from gene expression data integrated with curated biological

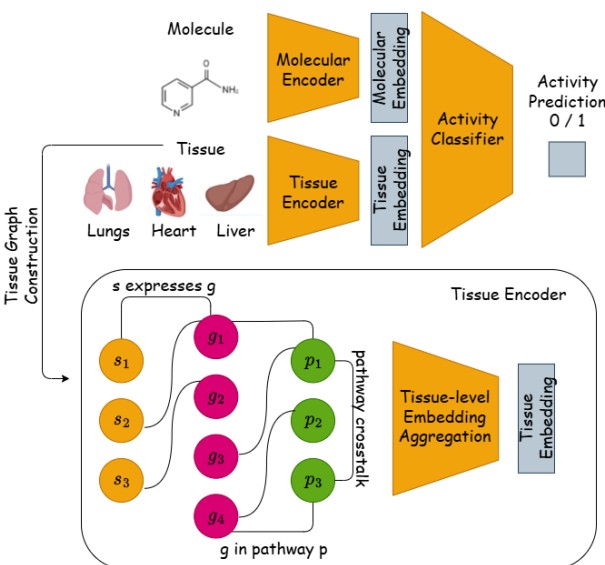

Figure 1: Overview of the Expresso model. A heterogeneous Sample–Gene–Pathway graph captures tissue-specific structure by integrating gene expression, gene–pathway links, and pathway–pathway crosstalk. Node types are color-coded: yellow for samples ($v_s$), pink for genes ($v_g$), and green for pathways ($v_p$).

knowledge. These two embeddings are concatenated and passed through a classifier to predict molecular activity.

We first pre-train the tissue encoder using self-supervised tasks—gene expression reconstruction, tissue classification, and pathway-level contrastive learning—across all tissues without molecular labels. During supervised training, the molecular encoder is frozen while the tissue encoder is fine-tuned. The combined loss, blending binary cross-entropy with self-supervised objectives, promotes biologically grounded and transferable tissue representations.

## 5 Experimental Results

In this section, we empirically evaluate the effectiveness of Expresso for tissue-specific molecular activity prediction. We first present the main results across nine human tissues, followed by an in-depth ablation analysis to dissect the contribution of each model component.

### 5.1 Main Result: Activity Prediction

We benchmark Expresso against six representative baselines spanning language-based, graph-based, and cross-modal paradigms. Full implementation and fine-tuning details for each baseline are provided in Appendix A.3. Additional evaluation metrics and extended results are provided in Appendix B. Table 1 summarizes the mean AUC scores per tissue. Expresso consistently outperforms all baselines across every tissue, achieving an average AUC of 0.812. This corresponds to a 3.7% relative improvement over the best-performing baseline, BioBERT.

Transformer-based molecular and biochemical language models (BioBERT, Mol-BERT and MolFormer) consistently achieve strong AUCs in tissues where larger numbers of compound–tissue examples are available (e.g., Liver, Breast), suggesting that these pretrained text- and molecule-based representations capture broadly useful chemical–semantic features when sufficient fine-tuning data are present. Their performance, however, decreases in tissues with limited samples (e.g., Brain) or pronounced class imbalance (e.g., Cervix, Kidney), reflecting the reduced ability of purely sequence- or text-derived encoders to capture tissue-specific biological variation under data-scarce or imbalanced conditions.

Table 1: Mean performance of Expresso for predicting molecular activity in human tissues, measured by ROC-AUC. Results show Expresso and baseline models, averaged over 5 runs with different random seeds. Higher values indicate better accuracy.

| | Model | Brain | Breast | Cervix | Kidney | Liver | Lung | Ovary | Prostate | Skin |
|---|---|---|---|---|---|---|---|---|---|---|
| | Expresso | **0.6471** | **0.8233** | **0.9325** | **0.8764** | **0.8207** | **0.8763** | **0.7466** | **0.6953** | **0.8950** |
| Baselines | BioBERT | 0.6297 | 0.7867 | 0.8948 | 0.8340 | 0.7796 | 0.8526 | 0.7234 | 0.6848 | 0.8622 |
| | CLAMP | 0.5546 | 0.7849 | 0.8815 | 0.8038 | 0.7712 | 0.8411 | 0.6899 | 0.6547 | 0.8648 |
| | GROVER | 0.5844 | 0.6018 | 0.6060 | 0.6056 | 0.6062 | 0.6017 | 0.6020 | 0.6208 | 0.6017 |
| | KA-GNN | 0.6008 | 0.7538 | 0.8461 | 0.7878 | 0.7491 | 0.8242 | 0.6935 | 0.6681 | 0.8122 |
| | Mol-BERT | 0.5630 | 0.7760 | 0.8840 | 0.8324 | 0.7777 | 0.8582 | 0.7137 | 0.6753 | 0.8672 |
| | MolFormer | 0.5882 | 0.7886 | 0.8989 | 0.8356 | 0.7834 | 0.8458 | 0.7049 | 0.6588 | 0.8683 |
| Ablation | Mol-BERT Encoder | 0.6387 | 0.7654 | 0.8885 | 0.8323 | 0.7764 | 0.8534 | 0.7032 | 0.6635 | 0.8741 |
| | ChemGPT Encoder | 0.6465 | 0.5699 | 0.6056 | 0.5563 | 0.5762 | 0.5247 | 0.5812 | 0.5854 | 0.5866 |
| | No Tissue Encoder | 0.5756 | 0.7871 | 0.8895 | 0.8061 | 0.7739 | 0.8173 | 0.6954 | 0.6781 | 0.8413 |
| | Non-Graph Encoder | 0.5672 | 0.7855 | 0.8884 | 0.8291 | 0.7801 | 0.8321 | 0.7149 | 0.6509 | 0.8480 |
| | One-Hot Encoder | 0.6076 | 0.8084 | 0.8962 | 0.8534 | 0.7885 | 0.8568 | 0.7425 | 0.6862 | 0.8649 |
| | Text-Based Encoder | 0.6092 | 0.8007 | 0.8993 | 0.8312 | 0.7849 | 0.8499 | 0.7222 | 0.6994 | 0.8718 |
| | Mean Tissue Expression | 0.6176 | 0.7891 | 0.8886 | 0.8325 | 0.7812 | 0.8416 | 0.7100 | 0.6718 | 0.8666 |
| | No Pretraining | 0.6180 | 0.8001 | 0.9110 | 0.8552 | 0.7924 | 0.8579 | 0.7242 | 0.6759 | 0.8663 |
| | Only $\mathcal{L}_{activity}$ | 0.6176 | 0.8097 | 0.8941 | 0.8493 | 0.7938 | 0.8557 | 0.7147 | 0.6707 | 0.8743 |
| | $\mathcal{L}_{activity} + \mathcal{L}_{gene}$ | 0.6361 | 0.8209 | 0.9106 | 0.8672 | 0.8129 | 0.8695 | 0.7322 | 0.6738 | 0.8812 |
| | $\mathcal{L}_{activity} + \mathcal{L}_{cls}$ | 0.6134 | 0.8126 | 0.9074 | 0.8481 | 0.7941 | 0.8592 | 0.7132 | 0.6733 | 0.8743 |
| | $\mathcal{L}_{activity} + \mathcal{L}_{pathway}$ | 0.6218 | 0.8141 | 0.9112 | 0.8553 | 0.8052 | 0.8695 | 0.7200 | 0.6786 | 0.8731 |
| | Only Sample Nodes | 0.5966 | 0.7876 | 0.8979 | 0.8208 | 0.7801 | 0.8370 | 0.7016 | 0.6803 | 0.8578 |
| | No Gene Nodes | 0.6050 | 0.8220 | 0.9054 | 0.8578 | 0.7998 | 0.8626 | 0.7372 | 0.6717 | 0.8776 |
| | No Pathway Nodes | 0.6387 | 0.8091 | 0.9044 | 0.8616 | 0.7949 | 0.8596 | 0.7328 | 0.6559 | 0.8733 |

Graph-based methods (GROVER, KA-GNN) and the cross-modal baseline (CLAMP) display complementary limitations that explain their comparatively lower performance. Graph encoders effectively capture molecular topology and local substructures but lack mechanisms to integrate tissue- or sample-level transcriptional information; consequently, they struggle to model differential compound activity driven by tissue-specific molecular mechanisms. CLAMP's multimodal design partially mitigates this by coupling molecular and textual embeddings, yet its contrastive fusion does not fully leverage expression- or pathway-level structure—leading to weaker biological grounding.

In contrast, Expresso explicitly integrates molecular structure, sample-level gene expression, and pathway topology, enabling context-aware molecular representations that better capture tissue-contextual patterns of compound activity. This biologically grounded integration yields more accurate and generalizable predictions across diverse human tissues.

## 5.2 Ablation Experiments

To understand the contributions of Expresso's design components, we conduct an ablation study focused on three key aspects: the encoders architecture, self-supervised objectives, and the biological graph structure.

### 5.2.1 The Role of the Molecular Encoder

This experiment examined how the choice of molecular encoder influences downstream performance. While Expresso employs the CLAMP (Seidl et al., 2023) molecular encoder, we replaced it with two alternatives to test different representation strategies: Mol-BERT (Li & Jiang, 2021), a transformer trained on SMILES strings, and ChemGPT (Frey et al., 2023), a generative model trained on textual molecular data. Mol-BERT performs reasonably well, showing that SMILES-based pretraining captures relevant chemical semantics, whereas ChemGPT underperforms, suggesting that textual generative models lack sufficient structural precision for accurate molecular activity prediction.

### 5.2.2 The Role of the Tissue Encoder

In this experiment, we tested various tissue representation approaches to understand their impact on model performance. The *One-Hot Encoder* uses simple, learnable one-hot embeddings to represent tissues, while the *Text-Based Encoder* employs frozen CLAMP text embeddings to capture tissue-specific textual information, but without task-specific adaptation. The *Non-Graph Encoder* utilizes precomputed pathway activity profiles from ssGSEA, removing pathway–pathway interactions to capture broad functional trends. Additionally, we evaluated *Mean Tissue Expression*, where each tissue is represented by the average gene expression across all samples in that tissue, capturing only the tissue's general activity profile.

The intuition behind these variants was to evaluate how different levels of tissue context affect performance. The results show that Expresso consistently achieves the highest performance across all tissue, while removing the tissue encoder leads to a marked decline. The *One-Hot* and *Text-Based* variants partially recover this loss but remain below Expresso, showing that simple or frozen representations are insufficient to capture complex biological variability. The *Mean Tissue Expression* baseline performs comparably to these simplified variants, indicating that averaged transcriptomic signals provide only coarse tissue-level cues.

### 5.2.3 The Role of Pretraining

Without pretraining, the tissue encoder lacks the opportunity to learn how to represent tissues in a generalizable way. It is only optimized to support the downstream molecular activity prediction task, which limits its ability to model gene and pathway interactions more broadly. Pretraining provides this missing context by exposing the encoder to diverse tissues and self-supervised objectives, enabling it to build richer, biologically grounded representations independent of activity labels.

### 5.2.4 The Role of Auxiliary Losses

We also examine how different self-supervised objectives affect learning. Removing all auxiliary losses reduces performance across the board, confirming that the model benefits from more than just label supervision. Gene-level masking helps the model capture fine-grained expression dynamics, while sample classification reinforces broader tissue-level structure. Pathway-based objectives are especially useful in tissues where higher-order biological processes dominate, enabling the model to reason more effectively about drug activity.

### 5.2.5 The Role of Graph Structure

The architecture of the tissue graph plays a central role in Expresso's success. When only sample nodes are used, the model lacks meaningful biological grounding. Removing gene nodes, by dropping all sample–gene and gene–pathway edges, eliminates crucial expression-level detail, while excluding pathway nodes strips away functional hierarchy. The full graph - composed of samples, genes, and pathways - enables both bottom-up signal integration and top-down semantic guidance. Each node type contributes a complementary perspective, and removing any of them weakens the model's ability to reason about tissue-specific drug effects.

### 5.2.6 Synthesis of Ablation Findings

The ablation study highlights that not all components of Expresso contribute equally to performance: the tissue encoder and the biological graph structure are the most critical. Removing the tissue encoder leads to the largest and most consistent drop in ROC-AUC, demonstrating that molecular features alone are insufficient to capture tissue-specific activity. Similarly, disrupting the graph—by omitting gene or pathway nodes—substantially impairs performance, emphasizing the importance of modeling hierarchical interactions between samples, genes, and pathways rather than relying on flat or averaged representations. Auxiliary self-supervised losses also improve predictions, particularly in tissues with complex pathway activity, but their effect is secondary to the structural and contextual components. Overall, these results suggest that Expresso's predictive power arises primarily from the combination of chemically informed molecular embeddings with biologically grounded, graph-structured tissue context.

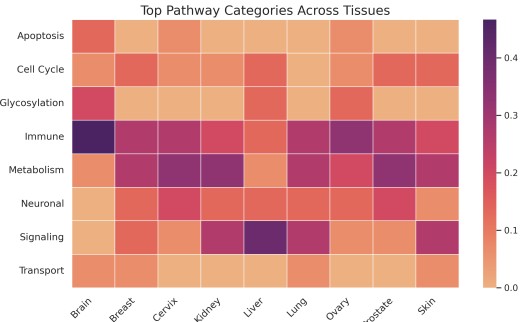

Figure 2: Heatmap of normalized pathway importance across tissues. Each row corresponds to a biological pathway grouped into functional categories, and each column represents a tissue. Colors indicate the relative contribution of each pathway to tissue-specific predictive performance, with darker colors representing pathways whose removal strongly reduces model accuracy, and lighter colors indicating pathways with minimal impact. Values are normalized per tissue such that 0 corresponds to no effect and 1 corresponds to maximal impact.

## 6 Uncovering Tissue-Specific Insights Through Pathways and Predictions

In this section, we present deeper insights into how the model captures tissue-specific drug behavior. We analyze the biological pathways it relies on across tissues, and evaluate its ability to prioritize therapeutically relevant compounds, offering both interpretability and predictive validation.

### 6.1 Pathway-Level Insights Across Tissues

To interpret Expresso's tissue-specific predictions, we conducted an interventional pathway ablation analysis. Pathways were grouped into functional categories by an expert. For each tissue, we systematically removed each pathway group from the model's internal graph and measured the resulting decline in predictive performance. The magnitude of this decline indicates the pathway group's relative contribution to prediction.

Figure 2 displays the normalized importance of pathway categories across tissues. Higher values indicate that removing a pathway category substantially reduces predictive accuracy, while lower values indicate minimal contribution. This visualization allows direct assessment of the relative functional dependencies captured by the model across tissues.

The pathway importance patterns reflect both general and tissue-preferential functional trends. Across most tissues, immune and metabolism pathways consistently contribute, highlighting their broad roles in defense and energetic homeostasis. Signaling pathways are particularly important in organs like the kidney and liver, consistent with the need for receptor-mediated regulation and intercellular communication. Neuronal pathways show limited contributions in the brain but moderate contributions in ovary and prostate, suggesting that tissue-specific neuronal dependency is not strongly captured by this metric. Apoptosis, cell cycle, glycosylation, and transport pathways show minor but consistent influence, representing baseline cellular maintenance.

Some tissue-preferential patterns are apparent but vary in magnitude: the brain emphasizes immune and maintenance-related pathways, reflecting immune–neuronal interactions; the liver relies on signaling and metabolism for detoxification and nutrient processing; the kidney shows strong signaling and metabolic importance, supporting filtration and homeostasis; and lung tissue balances metabolic and immune contributions, consistent with environmental exposure. Other tissues, including breast, cervix, ovary, prostate, and skin, display moderate contributions from multiple categories, reflecting a combination of biosynthetic, signaling, and defense processes. Overall, Expresso captures coherent, biologically meaningful pathway dependencies that vary across tissues while maintaining general functional consistency.

One might worry that ablated graphs are out-of-distribution and could lead to unreliable importance estimates (Hase et al., 2021); however, as shown in Appendix D, 91% of ablated graphs remain within the model's learned distribution, indicating minimal OOD impact.

Specific tissue-level observations mostly align with known biology. In the brain, immune signaling and apoptosis pathways, including FOXO-mediated transcription and complement cascades, contribute to neuronal maintenance and immune–neuronal interactions, while neuronal pathways have low apparent importance. Breast tissue emphasizes FGF and mTOR signaling, estrogen-regulated transcription, and glycosylation/mitochondrial pathways, reflecting hormone-driven growth and protein modification demand. The liver shows importance for signaling pathways such as FGFR1/2 and nitric oxide, as well as xenobiotic metabolism, lipid, and bile acid pathways, consistent with detoxification and metabolic regulation. Kidney pathways include heme scavenging, ion transport, and xenobiotic metabolism, matching renal clearance and electrolyte balance. Cervical tissue highlights metabolism, immune, and glycosylation pathways, supporting epithelial differentiation and local defense.

Overall, these results demonstrate that Expresso captures biologically meaningful tissue-specific functional dependencies, with some pathway contributions varying in magnitude across tissues, as reflected in the normalized heatmap.

## 6.2 Prioritizing Clinically Relevant Compounds

We evaluated tissue-specific prioritization by retrospectively testing FDA-approved drugs with known clinical indications. Each compound was input to the model across all tissues, and predicted binary activity scores were used to rank compounds within each tissue. This allows assessment of whether the model naturally assigns higher activity to drugs in the tissues where they are clinically effective.

The model successfully prioritized clinically relevant compounds. For instance, it highly ranked Sorafenib (Keating, 2017; Kane et al., 2006) in liver (0.7371), a therapy for liver cancer; Fulvestrant (Ciruelos et al., 2014; Nathan & Schmid, 2017) in breast (0.8126), an estrogen receptor antagonist used in breast cancer treatment; and Donepezil (Kertesz, 2004) in brain (0.9120), commonly prescribed for Alzheimer's disease. At the same time, it deprioritized compounds with limited outcomes, such as Talampanel, which had minimal efficacy in neurological conditions (0.3284).

These results, together with a more in-depth analysis of predicted tissue-specific activity provided in Appendix C, indicate that Expresso captures molecular signals aligned with known pharmacological effects across tissues. By integrating molecular structure with tissue-specific biological context, the model enables more targeted prioritization of compounds, bridging the gap between computational screening and experimental validation. This approach supports strategic experimental planning and enhances confidence in selecting compounds with higher potential for human relevance.

## 7 Conclusions

In this work, we presented Expresso, a multi-modal framework that enhances molecular activity prediction by incorporating tissue-specific biological context. While most existing models treat tissues as flat input, Expresso is designed to reason over the molecular and cellular environment in which compounds act, as represented by human cell-based in vitro assays. By embedding both molecular structure and rich tissue-level transcriptional information into a unified predictive model, Expresso narrows the gap between in vitro data and tissue-level biological settings, within the limits of the available assay panel.

We conduct an empirical evaluation across nine benchmarks involving thousands of molecules, demonstrating the value of leveraging tissue-specific biological information for activity prediction. Expresso consistently outperforms baseline architectures, with performance gains largely attributed to our novel tissue encoder. This encoder models gene–pathway interactions and captures sample-level variation within tissue types. By grounding the model in biological context, Expresso enables more accurate and generalizable predictions of compound activity across diverse human tissue contexts.

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
