# OpenReview forum: "Mechanism-Aware Prediction of Tissue-Specific Drug Activity via Multi-Modal Biological Graphs"
_TMLR — Accepted by TMLR_

### Review · Reviewer_LyGu · 2025-11-21

**Summary Of Contributions:**

This paper presents Expresso, a graph neural network for predicting drug activity in various tissues. Its unique value proposition is the incorporation of multimodal data within a multi-task framework to support the prediction: chemical structure of the compound/molecule, gene expression profile and biological pathways. Claims of outperformance relative to baselines are well-substantiated with detailed experiments and ablation studies. However, there is room for improvement in terms of the biological analysis is performed and reported. The manuscript will benefit from addressing the clarifying questions listed below (under ‘Requested changes’).

**Additional Comments:**

-

**Audience:**

Yes

**Audience Explanation:**

In the applied ML space, there are plenty of work on drug activity/response prediction and the use of graph neural networks for this purpose (customised for biological datasets) is an active area of research as it is well-known that baseline SOTA GNNs are insufficient.

**Broader Impact Concerns:**

Nothing of note for now.

**Claims And Evidence:**

Yes

**Claims Explanation:**

There are 3 main claims (1) mechanistically grounded representations, (2) outperformance relative to baselines, (3) new biological insights. Majority of them are supported with empirical evidence (though Claim 1 might be tougher to substantiate and could perhaps be rephrased or supported with additional evidence). However, the biological analysis requires some additional clarifications.

**Requested Changes:**

Major
1. Related work - this section is difficult to follow as everything is clustered into 1 paragraph, sentences jump around different topics and some papers are mentioned without linking back to how it is relevant to Expresso. A suggestion is to identify the key aspects of Expresso (e.g. multi-modal, multi-task), mention relevant papers related to each aspect (1 paragraph for each aspect) and always link back to how it matters (e.g. how is Expresso better).
2. Report the number of trainable parameters in each baseline model and Expresso.
3. Expresso’s sensitivity to model parameters, in particular the lambdas in the multi-task loss function, should be reported. (e.g. for each lambda, plot a chart of how model performance varies when lambda is varied from the existing optimised values)
4. Model explainability is done via a perturbation-based approach. This is well known to be confounded by an out-of-distribution (OOD) problem [1]. It would be good to investigate whether the knockouts are indeed OOD and if they are, how they impact the explanations/heatmap produced.
5. Section 6.2 is unclear as it does not seem to explain how exactly the testing was performed and how the ranks are computed.

Minor
- Keep the formatting consistent - subsubsections 4.2.1-4.2.4 have colons but not 4.3.2/3, 4.4.1… etc. Put it in all of them or remove from all.
- Comment on possible reasons why KA-GNN, a recent model published this year, has such consistently poor results across all datasets despite being the latest model out of all baselines.
- Add a subsection 5.2.6 to synthesise the findings from the ablation study. In particular, mention which aspects of the Expresso architecture were found to have the greatest impact on model performance and suggest why this might be the case.


[1] https://dl.acm.org/doi/10.5555/3540261.3540540

---

> ### Author Response · Authors · 2025-12-16
>
> Major:
> 1. Thank you for the suggestion on Related Work. We have extended and restructured the section, organizing it by key aspects of Expresso and clearly linking relevant papers to each.
> 2. The number of trainable parameters for Expresso and all baseline models has been added in Table 3 of the revised manuscript.
> 3. Thank you for the suggestion. Plots showing Expresso’s sensitivity to the multi-task loss weights have been added in Figure 3 of the revised manuscript.
> 4. We have verified that the perturbation-based knockouts are not out-of-distribution, and it is discussed in Appendix C.
> 5. Regarding section 6.2, we evaluated tissue-specific predictions by testing FDA-approved drugs with known clinical indications. Expresso ranked compounds higher in the tissues where they are clinically effective, demonstrating that the model captures tissue-specific molecular signals relevant to disease contexts.
>
> Minor:
> 1. As suggested, we have added Subsection 5.2.6 summarizing the ablation study findings, highlighting which components of Expresso most impact performance and providing possible explanations.
> 2. We added a short discussion in Section 5.1 about KA-GNN’s performance.

---

> > ### Comment · Reviewer_LyGu · 2025-12-19
> > **Did you upload the correct version of the revised manuscript?**
> >
> > Thank you for the response. While some points are addressed, I don't seem to be able to find Appendix C nor Table 3 in the revised manuscript?
> >
> > The latest PDF I can see consists of 15 pages, ending with Appendix A2. Could you check if you uploaded the correct version?

---

> ### Author Response · Authors · 2025-12-19
>
> Thank you for pointing this out.
>
> The submission is split as follows: the main paper (12 pages) together with the references appears in the primary PDF you are viewing (14 pages in total). Appendix C and Table 3 are not part of the main PDF; they are included in the supplementary material. The appendix spans pages 15–22 and is provided in the supplementary ZIP file, submitted as originally instructed.
>
> If this is not the preferred or correct submission format, please let us know and we will be happy to merge the appendix into the main manuscript and resubmit a unified version promptly.

---

> > ### Comment · Reviewer_LyGu · 2025-12-19
> > **A few more follow up questions**
> >
> > The main manuscript looks better / less confusing now, but you might want to remove the reference found on Page 1 in the supplementary materials. Perhaps using \newpage to divide the sections could do that.
> >
> > Below are some remaining concerns:
> > - Major point 3 - Noted that the figure has been added, but it will be better to write a sentence or two in the main manuscript about the key observations made from the sensitivity analysis in Figure 3.
> > - Major point 5 - ‘ranked higher’ is rather vague and potentially subjective. It would be good to provide more details of the activity scores and ranks produced by Expresso - Section 6.2 gives several examples of compounds and which organs are highly ranked, but presenting a table for 1 or 2 of these examples (showing scores and ranks) will be better to fully appreciate how well the prioritisation is done.
> > - Minor point 2 - I don’t seem to find a difference between the discussion in Section 5.1 in the original manuscript vs the revised one. Although the section does mention about KA-GNN, it was grouped together with GROVER as ‘graph-based methods’ and the discussion does not particularly mention why KA-GNN underperformed (even in comparison to GROVER). Now that we have Table 3 (number of trainable parameters in each model), we know that KA-GNN has much lower capacity as compared to other models and that could have been controlled better (e.g. increasing the number of nodes/layers such that the number of parameters across models are at least of the same order of magnitude). At this point, I don’t think this warrants additional experiments but it would be good to discuss about this in Section 5.1 (along with any other reasons you think might explain KA-GNN’s poor performance - particularly about KA-GNN and not just about graph-based methods).

---

> > > ### Author Response · Authors · 2025-12-19
> > >
> > > 1. We have added a brief discussion of the key observations from the sensitivity analysis in Figure 3 to the main manuscript (Section 4.4.5).
> > > 2. We have added specific ranking numbers in Section 6.2, showing the model’s predicted probabilities for each tissue, to make the prioritization results more transparent.
> > > 3. We re-ran the KA-GNN baseline with additional fine-tuning epochs and slightly increased the number of trainable parameters. As shown in Table 1, this improved its performance, so it no longer underperforms relative to the other baseline models. We can include a brief discussion specifically on KA-GNN if you still think it would be helpful.

---

> > > > ### Comment · Reviewer_LyGu · 2025-12-20
> > > >
> > > > 1. A sensitivity analysis reveals whether the presented results are reliant on careful tuning of hyperparameters (i.e. whether claims of outperformance are fragile), it does not 'show that incorporating each auxiliary loss consistently improves performance' - that is demonstrated via ablation studies. Please write the discussion for Figure 3 in a more careful manner.
> > > > 2. It would be more insightful to see more details, e.g. for 1 compound, scores for the various tissues tested and the ranks based on the score. Given that there is so much unknowns, I don't expect it to be perfectly in line with current knowledge/literature about the drugs but presenting these details would give more clarity wrt the strengths/weaknesses of the proposed method. Based on the current presentation, it is still not clear why 0.7371 for Sorafenib in liver is considered 'highly ranked' - what is the rank and what is this comparison relative to?
> > > > 3. What is the number of trainable parameters for KA-GNN now? You might need to update Table 4.

---

> > > > > ### Author Response · Authors · 2025-12-20
> > > > >
> > > > > 1. We thank the reviewer for this clarification. We have revised the discussion and added Appendix F, which focuses specifically on the sensitivity analysis in Figure 3 and carefully frames the results in terms of robustness to loss weighting.
> > > > > 2. We have added Appendix C, which provides the full predicted activity scores across all tested tissues along with an in-depth analysis.
> > > > > 3. Thank you for the observation. The updated number of trainable parameters for KA-GNN is now reported in Table 4.

---

### Review · Reviewer_TbHc · 2025-12-02

**Summary Of Contributions:**

The authors introduce Expresso, an architecture that incorporates biological context for predicting molecular activity in different tissues through heterogeneous graphs and multi-task pretraining. The downstream task is a binary classification task with molecule and target tissue pair inputs to predict molecular activity in that tissue. They compare their work across multiple baseline architectures (BioBERT, CLAMP, etc) and perform ablation studies (Section 5) and a knockout analysis across different biological pathways (Section 6). The motivation is relevant and the results consistently outperform baseline models. The authors provide documented code and describe their implementation details and hardware used for experimentation in the appendix. This work could benefit from some rewriting (mostly around citations). All in all, the work is of interest to the TMLR audience.

**Additional Comments:**

The submitted PDF files for the manuscript and appendix seem to prevent text selection. This made reviewing more difficult and time-consuming since I could not highlight the PDF for notes, search for terms in the document, and had to type out sections of text I wanted to highlight in my review. More importantly, it may impact accessibility and readability for readers (e.g. screen-reader users). Please ensure that the final submission is a text-selectable PDF.

**Audience:**

Yes

**Audience Explanation:**

Expresso's architecture and pretraining will be of interest to different downstream biological tasks, as the authors demonstrate the ability of their method to work across different tissue types.

**Claims And Evidence:**

Yes

**Claims Explanation:**

The main text contains sufficient explanation on the biological motivation and what the design decisions were (such as the compound encoder from CLAMP and respective sublosses). The Appendix contains sufficient information on data sources and splits (e.g. creating splits on the donor-level rather than sample level), includes additional metrics outside of AUROC, including Brier scores for calibration, and experiments to evaluate robustness to noise. The main text can benefit from more details on why certain design decisions were made (such as using MSE as a subloss, for example) while the code repository will benefit from scripts to run the experiments (more below for both points).

**Requested Changes:**

Paper comments:
- The default $\lambda$ hyperparameters for weighting sublosses in pretraining outlined in Section 4.4.5 seem to be 0.5, 0.5, and 0.25 for $\lambda_{\text{gene}}$, $\lambda_{\text{cls}}$, and $\lambda_{\text{pathway}}$, respectively. Were these the values used for experimentation? Did the authors perform any experiments to evaluate the impact of these values (outside of setting them to 0 for ablation)? In their ablation results, the $\lambda_{\text{activity}}+\lambda_{\text{gene}}$ results consistently outperforms the other ablated subloss models, and I’m curious if the weighting of the different $\lambda$ values may be part of the reason why.
- The ablations results demonstrate that sublosses are important for pretraining; however the decision on the specific functions used for the sublosses could use some justification. For example, the Gene Expression Reconstruction Loss in Section 4.4.1 is mean squared error over masked entries; it isn’t clear to me why this was chosen. Was it simply a simple and pragmatic choice, or is there a specific justification? A brief explanation or reference to the literature would help.
- In Appendix A.1 Implementation details, the authors discuss using “up to 15 fine-tuning epochs” for fine tuning–did they use early stopping, and if they did, under what metric?
- Table 4 in the Appendix shows a relatively high $p$-value for the DeLong test between the method (as well as for ovary and prostate) compared to very low $p$-values for the other organs. The authors comment on how low sample size may be the culprit: “...Expresso significantly outperforms the strongest baseline in most tissues, while a few tissues with smaller effect sizes did not reach statistical significance.” This is an interesting result and the authors should consider fleshing this out more.
- The code repository is well-documented. However, it would be nice if the authors included the experiment files in case anyone wants to recreate their ablation studies or knockout studies from Sections 5 and 6. The authors should also include the different settings they experimented on (e.g. the Appendix mentions that the models were trained on random seeds but the repository seems to only have the default seed).

Writing comments:
- The author names should be enclosed in in-text citations. For example, “typically by encoding it as a categorical label or an unstructured text input Mamoshina et al. (2016); Ingraham et al. (2019).” should be written as “(Mamoshina et al., 2016; Ingraham et al., 2019).” This might be an issue with \cite and \citet rather than \citep in LaTeX.
- Ensure that prior work is referenced correctly–for example Mol-BERT [1] is referenced as MolBERT.
- Some citations should be updated; for example, the Brody et al., 2021 paper “How attentive are graph attention networks?” [2] cites the arxiv preprint instead of the ICLR 2022 version.
- The captions for the figures and the tables could be fleshed out more such that they can be interpreted by themselves without referencing the main text (especially given that the PDF seems to not be searchable). For example, in Tables 1 and 2, the metric is AUC (I’m assuming it’s  AUROC), but there’s no mention in the caption, only in the first paragraph of Section 5.1. Or in the heatmap for Figure 2, how is a reader supposed to interpret the plot?
- Page 10, section 5.2.5 “The full graph— composed of samples, genes, and pathways—enables” has inconsistent spacing between words and em-dashes.

[1] Li, Juncai, and Xiaofei Jiang. "Mol‐BERT: An Effective Molecular Representation with BERT for Molecular Property Prediction." *Wireless Communications and Mobile Computing* 2021.1 (2021): 7181815.

[2] Brody, Shaked, Uri Alon, and Eran Yahav. 2022. “How Attentive Are Graph Attention Networks?” International Conference on Learning Representations.

---

> ### Author Response · Authors · 2025-12-16
>
> 1. Yes, the default subloss weights were used for all main experiments. We also explored the impact of varying these weights and included the corresponding performance graphs and we included the results in the appendix A.
> 2. The Gene Expression Reconstruction Loss (mean squared error over masked entries) was chosen as a standard, pragmatic choice for reconstructing masked gene expression values. Similar losses are commonly used in self-supervised learning for transcriptomic or molecular data, providing a straightforward way to encourage informative embeddings.
> 3. Early stopping was applied based on the validation loss, with training stopping if the loss did not improve for 5 epochs. Full details, including additional training settings, are provided in Appendix A1.
> 4. Thank you for pointing out the relatively high DeLong p-values for brain, ovary, and prostate. We have added a discussion in Appendix B on how sample size and baseline performance influence statistical significance in these tissues.
> 5. We will add the experiment files and all relevant settings, including different random seeds, to the GitHub repository to allow full reproduction of the ablation and knockout studies.
> 6. All writing-related comments have been addressed and the corresponding sections have been revised in the updated manuscript.

---

### Review · Reviewer_1TZT · 2025-12-03

**Summary Of Contributions:**

The authors present a model, Expresso, that predicts whether a drug has activity in a certain tissue, improving upon prior work that does not explicitly model tissues. The authors highlight their framework for learning tissue embeddings using graphs representing biological structure.

Strengths
1. The model formulation and architecture is clear and well-structured.
2. The setup of the tissue encoder using graphs built from GTEx data is fairly novel and interesting.
3. The authors conducted a comprehensive ablation study and benchmarked against several baselines, demonstrating superior performance.

Weaknesses
1. The proposed task may not be standard and/or well-defined. Indeed, neither the biological meaning of the binary targets (“tissue-specific drug activity”) nor how they are put together is well specified in the manuscript. Additionally, the toxicogenomics datasets (OPEN TG-GATEs and DrugMatrix) used in this paper are generally used for toxicity rather than therapeutic activity.
2. It is difficult to assess biological relevance and what the model is learning (Section 6) without clarity on the dataset, assay endpoints, thresholds, and sources of binary labels.
3. While Expresso shows moderate AUC improvements, results in the appendix on PR-AUC and leave-one-out validation are weaker but not discussed compared to baselines.
4. Key details on splits, hyperparameters, and uncertainty quantification are missing.

**Audience:**

Yes

**Audience Explanation:**

Although a combination of off-the-shelf GNN methods are used, the contribution is primarily in applying this to the tissue-specific activity prediction problem. This paper could be of interest to computational biologists, but the biological utility of prediction outputs is unclear.

**Broader Impact Concerns:**

NA, since publicly available data is used.

**Claims And Evidence:**

No

**Claims Explanation:**

The authors conducted clear benchmarks and ablation studies. However, more evidence or qualifications are required for these two claims:
Transferable/generalizable: Results in Table 6 need more discussion and specifications of splits across drugs and tissues.
Dataset and model interpretability: The biological endpoint is not clearly defined, and the model structure itself is not easily interpretable.

**Requested Changes:**

**Critical**

Introduction: The citations Mamoshina 2016 and Ingraham 2019 should be fixed as they are not related to prior work on similar tasks using simpler tissue encodings.

Related Work: The difference between what the cited works predict and the biological meaning of the task here needs to be clearly defined.

Appendix A2 (dataset description): “Activity labels…one of the most widely used and well-established benchmarks.” Please clarify or address all the points below:
- The type of experimental measurement and assay is not mentioned and not found in the papers cited or dataset link.
- The prior benchmarks using this dataset mentioned are not cited or compared to benchmarks in this work.
- The number of compounds is not mentioned anywhere in the paper aside from “thousands” (Section 7). The papers cited mention 600 and 170 compounds.
- The authors should further justify the choice of this toxicogenomics dataset for their task. Covariates such as dosage are not considered. The TG-GATEs dataset also seems to contain only human cancer cell lines, which are not necessarily aligned with GTEx data from human tissues.
- Many models for drug response prediction predict continuous outputs such as IC50 or dose-response curves. The authors should comment on the choice of supervised learning target, and how their work relates to previous efforts.

Text embedding baseline: Other methods (CLAMP) define bioactivity with a clear biological endpoint such as a bioassay or drug target. Clear descriptions of the specific bioassay generating a 0/1 output are passed to the text encoder. However, the BioBERT baseline in A.3 receives a vague prompt that does not define “molecular activity.” More details on BioBERT finetuning are also needed to evaluate the strength of the highest-performing baseline.

Metrics: The conclusions of the paper are too reliant on AUC (Table 1) compared to other metrics. More discussion on weaker PR-AUC and leave-one-out generalization (no longer outperforming baselines that do not use tissue embeddings) is needed. It would be good to see AUPRC for some of the baselines to determine if they are similarly affected by class imbalance.

Other details: The authors should mention more details about methods used for hyperparameter selection. It is also unclear how the 95% confidence intervals for ROC-AUC in Table 1 were constructed.

**Additional**

- Fig. 2 and Section 6.1 are confusing. The meaning of high/low score and color scheme should be clarified to make the text and figure consistent. For example, the text states “neuronal pathways are uniquely enriched in the brain” but Fig 2 shows normalized pathway activity of zero. Citations are missing.
- In Section 6.2, further justification is needed on how model predictions relate to specific diseases, since the GTEx dataset does not contain samples from those conditions.
- Important details about datasets, splits, other metrics should be in the main text.
- Sections 4-5 are thorough but can be shortened or partially moved to the appendix (e.g. repetition between 4.2.3 and 4.3.1).
- While statistically significant, Expresso’s performance gains are moderate. To further motivate the architecture, it could be interesting to look at how informative the context-aware sample embeddings or tissue embeddings are, perhaps through dimensionality reduction.

---

> ### Author Response · Authors · 2025-12-16
>
> General:
> 1. We clarified the biological meaning and construction of the binary activity labels. In our dataset, “activity” denotes whether a compound elicits a statistically significant response in a tissue-specific primary human cell assay (e.g., cytokine release in lung epithelial cells, transcription-factor activation in vascular co-cultures). These hit-calls are assigned by the EPA tcpl pipeline using standardized concentration–response modeling. We added explicit descriptions of these endpoints, the assay types per tissue, and the binary hit-call criteria in Section 3 and Appendix A2 of the revised manuscript.
> 2. In the revised manuscript, we rewrote Section 6 to more clearly describe the evaluation setting and experimental procedure. We now explicitly frame this analysis as a retrospective tissue-specific prioritization task using FDA-approved drugs with known clinical indications, where predicted binary activity scores are used for within-tissue ranking, rather than as assay-level endpoint prediction.
> 3. In the revised manuscript, we added the PR-AUC results for all baseline models in the Appendix (Table 5), along with a detailed discussion of the observed performance differences (Appendix B). We did not include leave-one-tissue-out (LOTO) experiments for the baselines. Conducting LOTO evaluation would require retraining each baseline separately for every tissue, resulting in a substantial computational overhead. Given that we already report two complementary evaluation metrics—ROC-AUC and PR-AUC—across all tissues, we believe the current evaluation sufficiently captures baseline performance and supports a fair comparison.
> 4. Full details on the data splits and hyperparameter settings are provided in Appendix A of the revised manuscript.
>
>
> Major:
> 1. We corrected the citation issues in the Introduction and clarified the related work.
> 2. Thank you for the suggestion on Related Work. We have extended and restructured the section.
> 3. Thank you for highlighting the specific points regarding dataset description. We have addressed all the dataset-related points in the final manuscript by expanding Appendix A2.
> 4. Appendix A3 now includes the exact prompts used, which define activity in terms of the specific tissue assay (e.g., cytokine release in lung cells), as well as full fine-tuning details for BioBERT. This clarifies how the baseline was constructed.
> 5. In the revised manuscript, we added the PR-AUC results for all baseline models in the Appendix (Table 5), along with a detailed discussion of the observed performance differences (Appendix B). We did not include leave-one-tissue-out (LOTO) experiments for the baselines.
> 6. We added a dedicated section in the appendix describing hyperparameter selection, including search ranges and performance graphs. The 95% confidence intervals for ROC-AUC were computed over 5 runs with different random seeds. All these details will be included in the final manuscript.
>
> Minor:
> 1. In the revised manuscript, we rewrote Section 6.1 and added a more informative description of Fig. 2, clarifying the experimental setup, the meaning of high/low scores, and the color scheme. We also expanded the discussion to more clearly interpret the results and pathway enrichments, ensuring consistency between the text and the figure.
> 2. Regarding section 6.2, we evaluated tissue-specific predictions by testing FDA-approved drugs with known clinical indications. Expresso ranked compounds higher in the tissues where they are clinically effective, demonstrating that the model captures tissue-specific molecular signals relevant to disease contexts.
> 3. The definition of activity and the meaning of binary labels are described in Section 3 of the main text. Additional dataset details, including splits and supplementary metrics, are provided in Appendix A2, which we now explicitly reference from the main text.
> 4. We have performed extensive ablation experiments (Section 5.2) to quantify the information captured by the context-aware sample and tissue embeddings. Additionally, we included a new analysis showing that ablated graphs are largely in-distribution (Appendix C), demonstrating that the embeddings meaningfully contribute to the model’s insights.

---

> > ### Comment · Reviewer_1TZT · 2025-12-31
> > **Follow up questions**
> >
> > The addition of PR-AUC metrics for baselines and dataset details are a lot clearer now. A few things could be clarified:
> >
> > - What does Up/Down refer to in Table 3?
> > - How many distinct compounds are used in the dataset?
> >
> > Based on the dataset description provided, I had a few follow up questions:
> >
> > 1. Table 3 shows that most of the assays use cancer cell lines, which may have very different physiology from healthy tissues in GTEX data.
> > This is mentioned as an issue in previous work (“cancer cell lines…struggle to generalize to normal human tissue,” pg. 2), but not assessed here.
> > Since each tissue label in the dataset corresponds to a single assay and single cell type or cell line, it would be good to justify why this generalizes to tissue activity for therapeutic applications.
> >
> >
> > 2. Section 6 is clearer now, but the discussion of pathway importance by tissue in Figure 2 does not fully back up claims of tissue-specific patterns. More careful discussion is needed.
> > For example, neuronal pathways have low importance in the brain in Fig. 2. The assay labeled as brain tissue only measures receptor binding on isolated membranes.
> >
> > Minor comments:
> >
> > - On page 3: “activity indicates whether a compound significantly affected a tissue-specific, biologically defined endpoint in a primary human cell assay.”
> > As already discussed in Appendix A2, many of the assays use cell lines rather than primary human cells, so this should be clarified in the main text.
> >
> > - On page 16: “Previous work has used subsets of these assays (e.g., BSK and APR from Open TG-GATEs and DrugMatrix) for similar classification tasks.”
> > What are BSK and APR, and what was the task?

---

> > > ### Author Response · Authors · 2025-12-31
> > >
> > > General:
> > > 1. In Table 3, Up and Down denote the direction of the assay signal relative to the control condition used in the original experiments, with Up indicating increased signal and Down indicating decreased signal. We have clarified this in Appendix A2.
> > > 2. The number of distinct compounds varies substantially across assays, ranging from tens to many thousands, depending on the specific experiment and data source. Table 3 presents a representative example of one assay per tissue, while the full dataset includes multiple assays and endpoints for each tissue, each with its own compound coverage.
> > >
> > > Major:
> > > 1.  We have slightly revised our claims, as reflected throughout the manuscript and in the Limitations section (Appendix I). Specifically, we now emphasize that our goal is to predict assay-defined tissue-contextual activity given the available panel, without claiming full extrapolation to healthy tissue physiology.
> > > 2. We fully agree that immortalized and cancer-derived cell lines differ from healthy tissues as profiled in GTEx, and that this limits how far one can extrapolate to in vivo physiology and clinical outcomes.
> > >
> > > Our intent in this work is not to claim that we directly model the full physiology of normal tissues or predict clinical responses in patients. Instead, our prediction targets are assay- and cell-type–specific tissue activities as they are operationally defined in a large human in vitro pharmacology panel, where each “tissue” corresponds to a specific cell-based assay in a particular human-derived cell line (often cancer-derived, with some primary cell systems). This follows standard practice in preclinical drug discovery, where tissue-of-origin cell systems are used as practical surrogates for tissue context in early-stage screening, safety flagging, and mechanism-of-action studies.
> > >
> > > To make this scope explicit and address the reviewer’s concern, we have revised the manuscript in several places. While we acknowledge that cancer cell lines are imperfect models of normal tissues, our focus is on the realistic preclinical setting in which tissue-of-origin cell systems define tissue context. The revised text clarifies this operational definition of “tissue,” narrows the scope of our claims to assay- and cell-type–specific tissue activity in vitro, and explicitly discusses the limitations regarding generalization to healthy GTEx tissues and therapeutic applications in patients.
> > >
> > > Minor:
> > > 1. We have added a clarification in Section 3.
> > > 2. We have clarified this point in Appendix A2.

---

### Decision · Action_Editor_K5qi · 2026-01-05

**Recommendation:** Accept with minor revision

**Additional Comments:**

A minor revision is requested to address a point raised during review. Please add a concise summary of the dataset composition, reporting for each tissue the number of assays and the number of distinct compounds.

**Audience:**

Yes

**Audience Explanation:**

Computational biologists and ML researchers working on drug response, toxicogenomics, and multimodal biological learning are likely to find the methodology and empirical findings informative, even where biological scope is appropriately limited to in vitro assay settings.

**Claims And Evidence:**

Yes

**Claims Explanation:**

The revised manuscript’s main claims are supported by clear empirical evidence. After clarification and revision, the authors define the prediction target and label construction (assay-defined binary “activity” hit-calls) and provide sufficient dataset/split and training details to evaluate the results. Across nine tissues, Expresso is compared against multiple baselines and supported by ablations, with consistent improvements in ROC-AUC.

---

> ### Author Response · Authors · 2026-01-15
>
> We have incorporated the requested clarification and have submitted the final revised version of the manuscript.